# The Role of MRI-Guided Radiotherapy for Soft Tissue Sarcomas

**DOI:** 10.3390/jcm11041042

**Published:** 2022-02-17

**Authors:** Grace C. Blitzer, Poonam Yadav, Zachary S. Morris

**Affiliations:** 1Department of Human Oncology, School of Medicine and Public Health, University of Wisconsin, Madison, WI 53726, USA; zmorris@humonc.wisc.edu; 2Department of Radiation Oncology, Northwestern Feinberg School of Medicine, Chicago, IL 60611, USA; poonam.yadav@nm.org

**Keywords:** MRI-guided radiation, soft tissue sarcoma, hypofractionated radiation

## Abstract

Soft tissue sarcomas (STS) are a rare class of tumors that originate from mesenchymal tissues and occur most frequently in the extremities, trunk, and retroperitoneum. Surgical resection with R0 margins is the primary curative treatment for most localized STS. In this setting, radiation therapy is used either pre-operatively or post-operatively to reduce the rate of local recurrence. Modern pre- or post-operative radiation therapy rely on the use of MRI sequences to guide target delineation during treatment planning. MRI-guided radiotherapy also offers unique advantages over CT-guided approaches in differentiating STS from surrounding normal soft tissues and enabling better identification of target volumes on daily imaging. For patients with unresectable STS, radiation therapy may offer the best chance for local tumor control. However, most STS are relatively radioresistant with modest rates of local control achieved using conventionally fractionated radiation. Specialized techniques such as hypofractionated radiation may allow for dose intensification and may increase rates of local control for STS. In these settings, MRI becomes even more critical for the delineation of targets and organs at risk and management of tumor and organ at risk motion during and between radiotherapy treatment fractions.

## 1. Role of Radiation for Soft Tissue Sarcomas

Soft tissue sarcomas (STS) are a heterogenous group of malignancies that arise from tissues of mesenchymal cell origin. STS are rare, accounting for <1% of all adult malignancies, with approximately 13,000 adults being diagnosed in the U.S. in 2021 [1]. There are more than 50 subtypes of STS, with each representing a unique clinicopathologic histology, with significant variation in radiosensitivity [2]. Surgical resection is a critical component of curative treatment for localized STS. In these settings, radiation therapy is commonly used together with surgery either pre- or post-operatively to reduce the rates of local recurrence [3,4].

Radiation therapy’s primary role in STS is to improve local control in conjunction with surgery. In general, radiation is recommended to be part of the treatment regimen of patients at high risk of local recurrence, such as in large, high-grade tumors or those with a concern for positive margins after surgery [5]. Pre-operative radiation fields are generally smaller with a more readily defined target in conventional computed tomography (CT) imaging compared to post-operative fields. Pre-operative radiation is also typically lower dose than post-operative radiation (50 Gy versus 60–66 Gy), with dose escalated in the latter to account for the potential reduced radiation sensitivity of tumor cells in the hypoxic setting of a surgically devascularized tumor resection bed [2]. Additionally, pre-operative radiation may reduce STS tumor size and/or thicken the surrounding pseudo-capsule, potentially allowing easier negative-margin surgical resection [6,7,8]. Given the above factors, many experts and consensus guidelines favor pre-operative radiation when feasible for STS patients. Magnetic resonance imaging (MRI) is a key component of radiation therapy contouring and planning, as will be discussed in the following section.

For patients with unresectable STS, radiation therapy may offer best chance for local tumor control. Conventionally fractionated radiation alone has been shown to provide approximately 45% and 33% local control for STS at 2 and 5 years, respectively, with higher doses associated with better overall survival and local control [9]. Recent studies suggest that delivering higher dose per fraction may provide excellent local control of soft tissue sarcomas [10,11,12]. With high dose per fraction approaches target delineation and motion management are crucial in order to limit dose to surrounding organs at risk (OARs). Consequently, as the role of SBRT grows in the treatment of STS, MRI guidance will likely become increasingly critical.

MRI guidance of radiation therapy is a new development over the past decade, with many investigations ongoing, including patients with STS [13]. MRI-guided radiation therapy provides the opportunity for improved target delineation and alignment, as well as motion management for organs at risk such as bowel. Additionally, MRI-guided radiation allows for adaptive approaches, modifying the radiation plan to account for inter-fraction changes in anatomy. In the future radiomics may allow for adaptive treatment plans based on predictive imaging characteristics [13].

## 2. Current Role of MRI in Planning Radiation Therapy for STS Patients

In pathology specimens, STS tumors commonly exhibit microscopic disease extending outside of the gross tumor. In one series, infiltrating tumor cells were found up to 4 cm from the pseudo-capsule surrounding STS tumors in 10 out of the 15 patients [14]. Interestingly, these infiltrating tumor cells were in areas with T2 enhancement on MRI in 9 out of 10 patients [14]. The histologies of STS exhibiting infiltrative cells can vary, and include myxofibrosarcoma and undifferentiated pleomorphic sarcoma, with tumor infiltrates that can be seen on MRI [15,16]. A phase II study, RTOG 0630, evaluated local control and toxicity following pre-operative intensity modulated radiation therapy (IMRT) for STS using reduced target volumes. RTOG 0630 found that IMRT using MRI for target delineation allowed for excellent local control and reduced toxicity comparable to historical controls [17]. This demonstration of effective local control despite reduced target volume expansion may suggest a critical role of MRI target delineation in accurately defining the extent of STS tumors. Based on this, current cooperative group trials recommend the use of MRI for external beam radiation therapy treatment planning.

For radiotherapy image segmentation in treatment planning, the T1 post-contrast MRI sequence is typically used to contour gross disease in patients with STS [18]. Additionally, the T1 MRI sequences can assist in defining the anatomical compartment and its normal anatomic components including muscles, fascia, bone, and neurovascular structures [18]. Accurate clinical target volume (CTV) delineation is crucial to increase the likelihood of a margin-negative resection. To that end, T2 MRI sequences are most useful for CTV contouring. T2 sequences help delineate soft-tissue extension and peritumoral edema as most STS are hyperintense relative the surrounding soft tissues [19].

## 3. Future Role for MRI in Radiation for STS

In patients with unresectable STS, radiation is often the only option for local therapy. However, conventionally fractionated radiation alone may not be adequate to achieve local tumor control in many patients with unresectable STS [9]. Stereotactic body radiation therapy (SBRT), or hypofractionated radiation, provides the opportunity to escalate effective dose and potentially increase local control of unresectable STS [20]. SBRT can be used in the setting of unresectable disease as definitive therapy and can also be used for oligometastatic or metastatic disease. Two randomized trials by Gomez and Palma found that SBRT for local consolidative therapy improved overall survival in patients with oligometastatic cancers [21,22]. The benefit of surgery in controlling lung metastases and improving survival has been shown in STS [23,24]. The role of SBRT for oligometastatic STS appears to be both safe and effective [25,26]. Hypofractionated radiation for palliation may allow for improvement of patient quality of life and compared to conventionally fractioned approaches affords decreased treatment time and possibly an increased durability of palliative response. Consequently, SBRT and hypofractionated radiotherapy for STS is becoming more prevalent [27]. However, more research and follow up of hypofractionated radiation for STS is needed to evaluate the potential late toxicities. The use of MRI for target delineation is critical to these approaches because of the heightened need for target volume delineation and motion management with hypofractionation and dose intensification.

An expanding field of radiotherapy research is evaluating the role of MRI in quantitative imaging, providing information on tumor characteristics that may enable more personalized approaches to radiotherapy treatment planning and adaptation [13]. MRI can identify tumor hypoxia, as well as early response to radiation [13] and may thereby allow for precise dose-adaptation delivering higher or lower dose selectively to regions of a target volume. Additionally, MRI is being used in radiogenomics to define and correlate imaging phenotypes together with genomic characteristics of a tumor [28]. Such an approach may enable the use of MRI for personalizing radiation plans based on radiogenomic predictors of response [13]. This could be especially beneficial in the setting of a heterogeneous group of tumors such as STS, where generalized treatment paradigms are used despite the wide variation in histology, genomics, and radiotherapy response.

Heterogeneity within an individual STS tumor results in a lack of absolute resistance to radiation, which may allow for dose escalation, or biological equivalent dose escalation, to overcome relative radioresistance among a heterogeneous tumor. There is also a possible correlation between the heterogeneity seen on MRI with the biologic tumor heterogeneity. There is great potential for using MRI or functional imaging to determine radiographic correlates of radioresistance within specific areas of a heterogeneous tumor, with consideration for escalation of radiation dose to resistance-associated regions of a tumor.

## 4. Target Delineation and Motion Management Using MRI

Target delineation is crucial to the planning and delivery of radiation therapy. For STS, the target volumes generally consist of the gross disease (in pre-operative and definitive radiation) and areas concerning for microscopic spread of STS. MRI provides superior soft-tissue delineation and multiplanar capability. In addition to target delineation during the radiation contouring and planning process, target delineation during daily treatments is critical. With the use of IMRT and reduced geometric expansions, the need for precision in daily tumor alignment has increased [29]. Extremity STS are among the most challenging tumor sites to align for daily treatment due to the considerable mobility and the multi-axis degrees of freedom associated with these tumors. Molds and immobilization devices can begin to redress this but even with these tools, the use of daily image guidance is often needed to ensure correct tumor alignment during each fraction. Both cone beam CT and surface guidance technology can be used, and may allow for reduced planning target volume (PTV) expansions [30]. However, the ability of MRI to resolve STS tumors from surrounding normal soft tissues makes this a superior modality for daily image guidance of STS radiotherapy (Figure 1).

Accounting for the intra-fraction motion of tumor and surrounding OARs presents another challenge for STS. This is particularly salient for STS of the retroperitoneum and trunk, where many critical OARs continuously move and the tumor itself can be affected by respiratory and digestive motion. MRI-guided radiation therapy provides for real-time visualization of STS tumors and the surrounding OARs in these locations. This enables motion-gated delivery of radiation to tumor and minimizes dose to surrounding OARs.

## 5. Illustrative Cases on MRI-guided Radiation Therapy for Target Delineation and Motion Management

Here, we describe three cases of STS that were treated on a phase II clinical trial of hypofractionated radiotherapy for unresectable disease. Each patient had STS that was treated on an MRI-guided linear accelerator. One patient had STS in the upper extremity, one in the lower extremity, and one in the retroperitoneum. The patient with an unresectable upper extremity tumor had angiosarcoma histology with a target volume of 607 cubic centimeters (cc) and received 54 Gy in six fractions. The patient with an unresectable lower extremity tumor had dedifferentiated liposarcoma histology with a target volume was 2689 cc and received 48 Gy in six fractions. The patient with the unresectable retroperitoneal STS had spindle cell histology with a target volume of 81 cc and received 48 Gy in six fractions.

Daily MRI guidance allowed for the safe delivery of high dose hypofractionated radiation, with all patients having no grade 3 or higher acute toxicities. Understanding of potential late toxicity risks is still developing as clinical experience increases with this approach in patients with STS. For each of these patients, radiotherapy planning scans were acquired on 0.35T MRI (MRIdian, Viewray Inc., Mountain View, CA, USA) using a 3D True Fast Imaging (TRUFI) sequence, with 3 mm slice thickness and 40 cm × 40 cm × 43 cm field of view. A daily fractional MRI was acquired using the same imaging protocol to aid treatment set-up. In the case of the patient with STS of the lower extremity, MRI imaging enabled reproducibility of the positioning required to target the mobile lower extremity while avoiding the contralateral leg (Figure 1). In the case of the patient with STS of the upper extremity, MRI imaging ensured an accurate daily set-up and target delineation both for the planning and for each treatment (Figure 2). In the case of the patient with a retroperitoneal STS, daily MRI visualization allowed for adaptive planning which optimized tumor coverage while reducing dose to nearby organs at risk including bowel and kidney (Figure 3). Daily adaptation requires editing the OARs and/or PTV and re-optimizing the planned dose. Patients undergoing adaptive radiation are scheduled for 60 to 90 min. Based on the size of the lesion, editing contours may take approximately 10–20 min, re-optimization and plan review process may take additional 6–9 min.

In all cases, these large STS tumors were safely treated using hypofractionated radiation therapy. The patients with extremity STS had follow-up imaging demonstrating stable disease without local progression at 3 and 9 months for the cases of lower extremity STS and upper extremity STS, respectively. The patient with retroperitoneal STS had partial response on 6-month follow-up imaging. The use of MRI for daily set-up allowed for 0.3 mm PTV margins to be used for this patient, potentially decreasing the toxicity of the plan while affording confidence in tumor and OAR visualization and dosimetry.

## 6. Conclusions

STS is a challenging disease with over five thousand deaths estimated in the U.S. in 2021 [1]. There is a tremendous need for research into novel treatment techniques and therapies that can improve local control of STS patients while reducing treatment toxicities. Radiation therapy is crucial to the treatment of STS both in the definitive and palliative settings. Current radiation techniques depend on accurate contouring and positioning based on MRI. Future techniques may increasingly incorporate MRI for daily target delineation, motion management, dose adaptation, and radiogenomics.

## Figures and Tables

**Figure 1 jcm-11-01042-f001:**
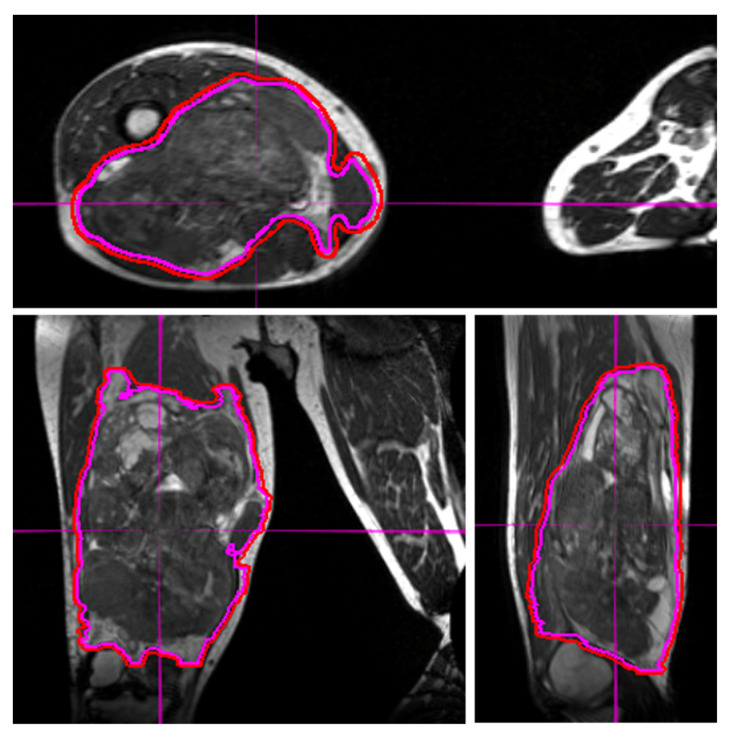
Visualization of an STS and normal tissues in the lower extremity using the 3D True Fast Imaging sequence on MRI-guided linear accelerator for alignment during treatment. The excellent delineation of the tumor and surrounding structures may allow for dose escalation and/or decreased toxicities. The use of daily MRI at set-up improves visualization and allows for smaller margins of expansion from gross tumor volume (magenta line) to planning treatment volume (red line).

**Figure 2 jcm-11-01042-f002:**
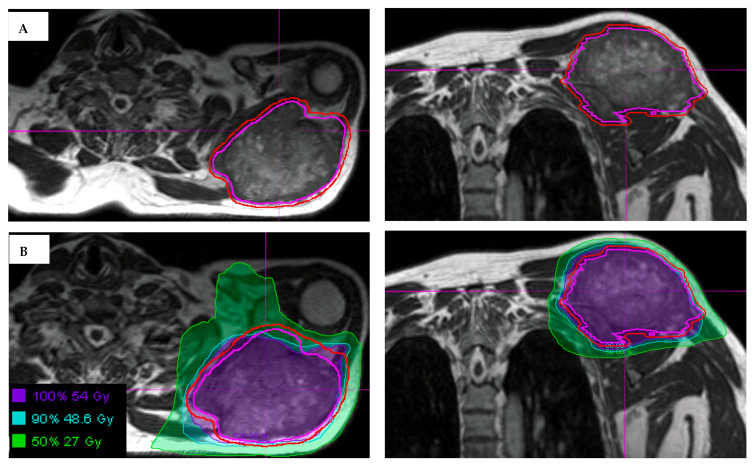
(**A**) Visualization of an STS and normal tissues in the upper extremity using the 3D True Fast Imaging (TRUFI) sequence on MRI-guided linear accelerator. The gross tumor volume is outlined in magenta, with expansion to planning treatment volume (outlined in red) using small margins, in part due to MRI for daily set-up. (**B**) Hypofractionated radiation treatment plan for an upper extremity STS showing the use of MRI-guidance to delineate the tumor and the surrounding at risk areas.

**Figure 3 jcm-11-01042-f003:**
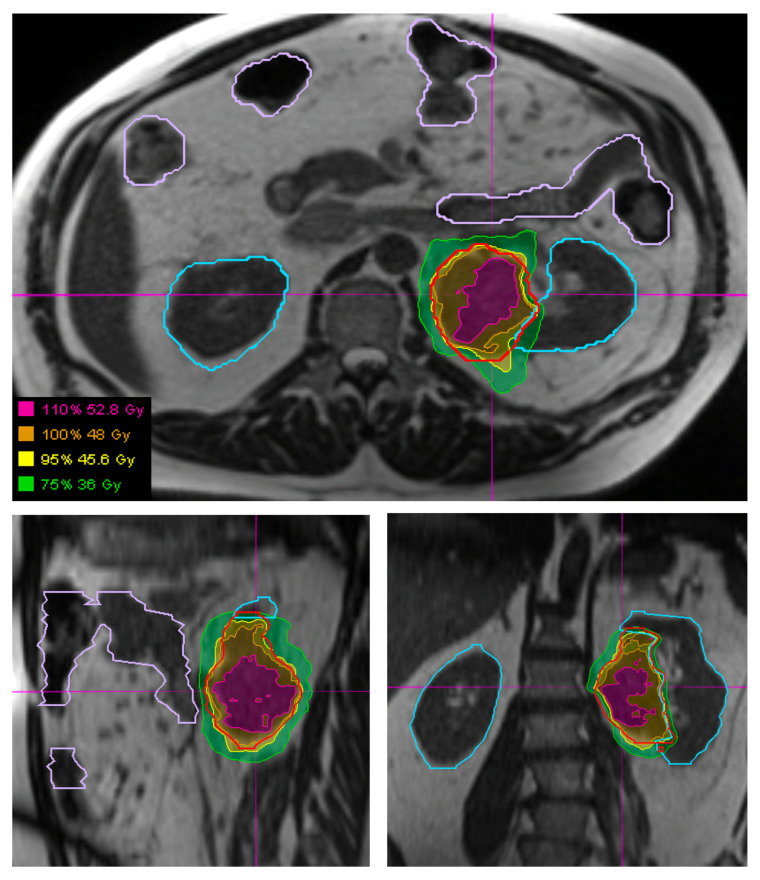
Visualization of a retroperitoneal STS (outlined in red) and surrounding organs at risk, specifically bowel (outlined in light purple) and kidneys (outlined in light blue), using the 3D True Fast Imaging (TRUFI) sequence on a MRI-guided linear accelerator. Hypofractionated radiation plan showing the plan which was adapted each fraction to optimize bowel and kidney sparing.

## Data Availability

Not applicable.

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
