# Peer review of "The Role of MRI-Guided Radiotherapy for Soft Tissue Sarcomas"

_jcm, 2022, doi:10.3390/jcm11041042_

Round 1

Reviewer 1 Report

This is an excellent manuscript which contributes to our knowledge regarding the importance of MRI guided imaging to improve accuracy in the therapy of soft tissue sarcomas. I would make the point that for hypofractionated RT, it is not possible to say if this therapy is not associated with significant morbidity unless long-term follow-up is undertaken in these patients to assess for complications such as soft tissue fibrosis, lymphedema, vascular damage and radionecrosis of bone.

Author Response

Reviewer 1: This is an excellent manuscript which contributes to our knowledge regarding the importance of MRI guided imaging to improve accuracy in the therapy of soft tissue sarcomas. I would make the point that for hypofractionated RT, it is not possible to say if this therapy is not associated with significant morbidity unless long-term follow-up is undertaken in these patients to assess for complications such as soft tissue fibrosis, lymphedema, vascular damage and radionecrosis of bone.

  • Excellent point. We have adjusted our manuscript to say “However, more research and follow up of hypofractionated radiation for STS is needed to evaluate the potential late toxicities.” In section 3 where hypofractionated radiation is discussed.
  • Additionally, we have added “understanding of potential late toxicity risks is still developing as clinical experience increases with this approach in patients with STS.” In section 5 where some case examples of hypofractionated radiation are presented.

Reviewer 2 Report

This is a review paper on MRI guided radiation therapy (RT)  in soft tissue sarcoma (STS). 

In the introduction I miss a remark about radiosensitivity of different STS subtypes, especially about non-radiosensitive entities.

1. Role of radiation therapy (RT) in STS:

How do the authors select patients for preoperative RT, do they use an app for predicting or do they use clinical criteria.

The term radiomics is only touched upon; it would be interesting to learn what exactly the term entails in view of the authors

2.Current Role of MRI in Planning Radiation Therapy for STS Patients 

Please specify infiltrating STS in more detail. Which histologies are particularly affected and which are not.

Please explain abbreviations once in the manuscript  e.g. CTV, PTV 

3. Future Role for MRI in Radiation for STS 

Please discuss briefly the advantages and disadvantages of SBRT versus surgery in patients with unresectable STS 

Please discuss the heterogeneity within a single tumor and the consequences thereof for your approach

4. Target Delineation and Motion Management using MRI 

Figure 1 could possibly be enlarged for better discrimination

Legend Fig 1.: the legend could be formulated in a more understandable way

It would be interesting to see an image at the beginning and at the end of RT to estimate the changes over time.
A remark about time required to daily  adjust radiation dose and - field would be useful.

5. Illustrative Cases on MRI-guided Radiation Therapy for Target Delineation and Motion Management 

You nicely describe three cases from your study. Histologies of these cases would be interesting, please add.

Additional remarks:

  • do you have experience with RT and concomitant systemic therapy and your adaptive technique ?
  • same question on systemic therapy followed by radiation therapy ?
  • role of machine learning approaches for your MRI technique ?

Author Response

In the introduction I miss a remark about radiosensitivity of different STS subtypes, especially about non-radiosensitive entities.

  • The variation in STS subtypes is a good point, and a great challenge to the treatment of STS. We have added the following statement to the introduction “There are more than 50 subtypes of STS, with each representing a unique clinicopathologic histology, with significant variation in radiosensitivity.”
  1. Role of radiation therapy (RT) in STS:

How do the authors select patients for preoperative RT, do they use an app for predicting or do they use clinical criteria.

  • At our institution all STS cases are discussed at our multidisciplinary tumor board and a consensus treatment plan is reached for each patient based on their individual clinical criteria. We have added the following sentence in section 1: “In general, radiation is recommended to be part of the treatment regimen of patients at high risk of local recurrence and/or those with limited salvage options in the event of recurrence. The timing of radiation relative to surgery is determined on a case-by-case basis after considering tumor location, risks for wound healing complications, and the acuity of the patient’s symptoms.”

The term radiomics is only touched upon; it would be interesting to learn what exactly the term entails in view of the authors

  • Radiomics is an interesting and rapidly expanding field. We believe that radiomics can be used to study the gene expression (and eventually aid in treatment selection) using imaging characteristics. This field is young however, and more research is needed prior to clinical implementation.

2.Current Role of MRI in Planning Radiation Therapy for STS Patients 

Please specify infiltrating STS in more detail. Which histologies are particularly affected and which are not.

  • Good question. The histologies can vary, and these margins are recommended for all STS. We have included the following sentence: The histologies of STS exhibiting infiltrative cells can vary, and include myxofibrosarcoma and undifferentiated pleomorphic sarcoma, with tumor infiltrates that can be seen on MRI.

Please explain abbreviations once in the manuscript  e.g. CTV, PTV 

  • All abbreviations have now been defined.
  1. Future Role for MRI in Radiation for STS 

Please discuss briefly the advantages and disadvantages of SBRT versus surgery in patients with unresectable STS 

  • We discuss all STS cases at our multidisciplinary tumor board and a consensus treatment plan is reached for each patient. In general, we would not recommend SBRT over surgical resection in cases where surgical resection would be curative as SBRT is still experimental. We do offer SBRT or hypofractionated radiation for STS that are localized and cannot be resected by our surgical oncologists. Additionally, SBRT allows for completion of radiation therapy more quickly with higher dose per fraction in low alpha/beta tumors that may be otherwise radioresistant, as compared to conventional radiation.

Please discuss the heterogeneity within a single tumor and the consequences thereof for your approach

  • We have added the following to the manuscript: “Heterogeneity within an individual STS tumor results in a lack of absolute resistance to radiation, which may allow for dose escalation or biological equivalent dose escalation to overcome relative radioresistance among a heterogeneous tumor. There is also a possible correlation between the heterogeneity seen on MRI with the biologic tumor heterogeneity. There is great potential for using MRI or functional imaging to determine radiographic correlates of radioresistance within specific areas of a heterogeneous tumor, with consideration for escalation of radiation dose to resistance-associated regions of a tumor.”
  1. Target Delineation and Motion Management using MRI 

Figure 1 could possibly be enlarged for better discrimination

  • Figure 1 has been enlarged.

Legend Fig 1.: the legend could be formulated in a more understandable way

  • The Figure legend has been changed to make it more understandable: “Figure 1. Visualization of an STS and normal tissues in the lower extremity using the 3D True Fast Imaging sequence on MRI-guided linear accelerator for alignment during treatment. The excellent delineation of the tumor and surrounding structures may allow for dose escalation and/or decreased toxicities. The use of daily MRI at set-up improves visualization and allows for smaller margins of expansion from gross tumor volume (magenta line) to planning treatment volume (red line). “

It would be interesting to see an image at the beginning and at the end of RT to estimate the changes over time.

  • In general, we do not see rapid shrinking of STS. Most of the STS treated at our institution with hypofractionated radiation are stable or have partial response. A large response is unusual, especially in the 2-3 weeks over which these treatments are generally given. Thus, we have not included this image as it would not show significant change over a short period of time.

A remark about time required to daily  adjust radiation dose and - field would be useful.

  • The role of adaptation is to limit dose to OARs and accelerate the dose to target on favorable anatomical days. We have added the following statements to the manuscript: “Daily adaptation requires editing the OARs and/or PTV and re-optimizing the planned dose. Patients undergoing adaptive radiation are scheduled for 60 to 90 minutes. Based on the size of the lesion editing contours may take approximately 10-20 minutes, re-optimization and plan review process may take additional 6-9 minutes.”
  1. Illustrative Cases on MRI-guided Radiation Therapy for Target Delineation and Motion Management 

You nicely describe three cases from your study. Histologies of these cases would be interesting, please add.

  • Histologies have been added in to section 5

Additional remarks:

  • do you have experience with RT and concomitant systemic therapy and your adaptive technique ?
    • We do not allow for systemic therapy with hypofractionated radiation. Our clinical trial has the use of systemic therapy as an exclusion criteria, all therapies must be stopped at least 2 weeks prior to trial enrollment.
  • same question on systemic therapy followed by radiation therapy ?
    • We do allow for hypofractionated adaptive radiation after systemic therapy on our clinical trial, as long as it has been stopped at least 2 weeks prior. However, as many STS do not respond well to systemic therapy this is not something we see for the majority of our patients.
  • role of machine learning approaches for your MRI technique ?
    • Excellent question. We are currently working with our medical physics group investigating this along with other MRI questions.

Reviewer 3 Report

Very interesting report on a very small series; larger series and longer follow up will be required to determine efficacy of present therapy and eventually update of adaptative planning.

Author Response

Reviewer 3: Very interesting report on a very small series; larger series and longer follow up will be required to determine efficacy of present therapy and eventually update of adaptative planning.

  • We agree that further research should be conducted into the effects of hypofractionated, adaptive radiation. We have added “Understanding of potential late toxicity risks is still developing as clinical experience increases with this approach in patients with STS.” In section 5 where some case examples of hypofractionated radiation are presented.

Round 2

Reviewer 2 Report

Dear Authors 

thank you for adequately adapt your MS according to the suggestions